# Factors That Influence the Sustainability of Human Milk Donation to Milk Banks: A Systematic Review

**DOI:** 10.3390/nu14245253

**Published:** 2022-12-09

**Authors:** Christelle Kaech, Catherine Kilgour, Céline J. Fischer Fumeaux, Claire de Labrusse, Tracy Humphrey

**Affiliations:** 1School of Nursing, Midwifery and Social Work, Faculty of Health and Behavioural Science, The University of Queensland, St Lucia, QLD 4072, Australia; 2HESAV School of Health Sciences, HES-SO University of Applied Sciences and Arts Western Switzerland, 1011 Lausanne, Switzerland; 3The Royal Brisbane and Women’s Hospital, Queensland Health, Herston, QLD 4029, Australia; 4Department of Woman-Mother-Child, Lausanne University Hospital, 1011 Lausanne, Switzerland; 5Faculty of Biology and Medicine, University of Lausanne, 1015 Lausanne, Switzerland

**Keywords:** donor human milk, donation sustainability, human milk banks

## Abstract

Donor human milk is the recommended alternative for feeding preterm or low birth weight infants when the mother’s own milk is unavailable or not in sufficient quantity. Globally, the needs of vulnerable infants for donor human milk exceed the supply. This review aimed to identify the factors impacting the sustainability of human milk donation to milk banks. A systematic review of the literature was performed on eight databases to retrieve articles published until December 2021. The study protocol is available in PROSPERO (#CRD42021287087). Among the 6722 references identified, 10 studies (eight quantitative observational and two qualitative) met the eligibility criteria for a total of 7053 participants. Thirty factors influencing the sustainability of the donations to milk banks were identified and categorized as follows: (1) donation duration, (2) donors’ infant features (e.g., gestational age, birth weight), (3) donors’ features (e.g., socio-demographic characteristics, milk donation history), and (4) factors related to the milk bank and health care systems (awareness and support). The available evidence suggests that larger volumes of donated milk are associated with a longer duration of donation, as are early donation, previous milk donation, and donors with an infant of smaller weight and gestational age. Supporting and encouraging early donation and recruiting donors with infants of low birth weight and low gestational age could support longer donation times and greater volumes of milk donated. To identify efficient strategies and to draw appropriate recommendations to improve donor milk access, future studies should further explore the issues of the sustainability of human milk donation to milk banks.

## 1. Introduction

A mother’s milk is the optimal nutrition for infants. Helping mothers sustain their milk supply should always be a priority for healthcare services. Breastfeeding support should be particularly emphasised for vulnerable infants, such as preterm or low-birthweight infants, as their mother’s own milk provides them with additional protection. However, matching supply with demand can initially be difficult for mothers [1,2]. If an adequate supply is unavailable, the second best source of nutrition for vulnerable infants is donor human milk (DHM) [1,3]. Human milk banks (HMBs) are responsible for providing safe DHM. They select and qualify donors and collect, treat (generally by pasteurisation), store, and deliver DHM.

Worldwide, approximately 15 million premature infants (<37 weeks of gestation) are born annually [4]. This population is at a greater risk of severe complications, such as necrotising enterocolitis (NEC) or sepsis [5,6]. NEC affects approximately between 2 and 22% of premature infants born before 32 weeks of gestation, with a mortality rate from 21.9 to 38.0% and high rates of short- and long-term morbidity among survivors [7,8]. The risk of NEC is reduced by half for DHM-fed vulnerable infants compared to formula-fed vulnerable infants [9]. Despite the increased number of HMBs globally, the supply of DHM does not meet infant demand in many countries [5,10,11,12,13]. Worldwide, more than 800,000 vulnerable infants received DHM from HMBs in 2020, but approximately 500,000 infants of less than 32 weeks lacked access to DHM [13]. HMBs need efficient strategies to enhance the DHM supply. Improving the sustainability of milk donation is one intervention that can help HMBs improve access to DHM, so better understanding the factors affecting the sustainability of donations to HMBs is a public health urgency.

In 2019, two systematic reviews investigated individual donors’ characteristics, motivations, and barriers/facilitators influencing their donations to HMBs, with findings categorised as an individual (donor), systemic (health care system), and social factors [14,15]. Both systematic reviews assessed the methodological quality of the included studies, but neither reported the confidence level in its findings or described the limitations of the review.

The present review acknowledges the essential role of donors and explores how to maintain regular donations, which is crucial to long-term sustainability. Donation sustainability was identified and evaluated through factors related to the volume of milk donation and the frequency or duration of donations, as well as the factors that contribute to retaining donors. Increasing the number of new donors is another option for improving the sustainability of donations. However, it is important to keep in mind that the recruitment process, donor screening, serology testing, and milk transportation are costly. For example, the logistical and economic burden is not equivalent between 20 women who will each donate 10 litres and 200 women who will each donate one litre. As reported by the PATH, sustainability requires not only a balanced DHM supply and demand but also financial sustainability [5].

This review does not address the factors influencing donor recruitment, why donors choose to donate, or why some women choose not to donate, as these have already been investigated [14,15]. Instead, this review advances knowledge of the factors that promote the sustainability and duration of milk donations to HMBs. To the best of our knowledge, no systematic reviews have addressed the factors influencing the sustainability of donation or the donors’ willingness and/or ability to continue donating milk.

The donor literature has focused mainly on blood donation sustainability, which has been widely investigated with large pools of donors [16,17,18]. The literature on milk donation sustainability is extremely scarce, and milk donation faces different challenges. For instance, only lactating individuals can donate, and they are a specific segment of the population during a specific, highly variable window of time. Research-based strategies must be found to sustain milk donations.

This review aimed to assess and synthesise contemporary evidence on the factors that sustain the donation of human milk to not-for-profit HMBs by existing lactating donors. The factors influencing donation sustainability are explored at the micro (individual, donor), meso (institutional), and macro (systemic) levels and include donor characteristics and behaviours, professional practices, organisational structures and donation practices, guidelines, regulations, and policy.

## 2. Materials and Methods

The protocol of this systematic literature review was published in the international prospective register of systematic reviews (PROSPERO) (#CRD42021287087). Two reference methods were followed when reporting this review: preferred reporting items for systematic reviews and meta-analyses (PRISMA) [19] and enhancing transparency in reporting the synthesis of qualitative research (ENTREQ) [20]. The completed PRISMA checklist is available in Appendix A.

### 2.1. Search Strategy and Selection Criteria

The first author developed the literature search strategy for this mixed-methods systematic review, which was subsequently peer-reviewed by the co-authors as well as two librarians from universities in Australia and Switzerland. MEDLINE, CINAHL, EMBASE, Cochrane Library, CENTRAL, SCOPUS, PsycINFO, and Web of Science were systematically searched for quantitative mixed-methods and qualitative studies published between the databases’ inceptions before 31 December 2021. The search was first performed in November 2021 and was updated in January 2022 (before the data analysis) to ensure that the most up-to-date literature was included. The main terms that were sought included ‘milk bank’, ‘human milk’, ‘donor milk’, ‘donation’, and ‘sustainability’. Further details on the literature search strategy are provided in Appendix A. The search was completed by manually searching the reference lists of relevant papers and guidelines, performing a grey literature search on Google Scholar, and conducting specific author and journal searches.

The inclusion criteria were primary research studies written in English or French that investigated factors influencing the sustainability of human milk donation to HMBs. The study population comprised human milk donors or any stakeholders involved in HMBs. The exclusion criteria were studies (1) for which the methods used were unclear; (2) with no full article available; (3) that focused exclusively on milk sharing for-profit milk banks or human milk sales or (4) that were published before 1990, as the phenomena of interest for those studies were not relevant to this review. The papers yielded by the database searches were uploaded to Covidence for the automated removal of duplicates. The first two authors screened all the articles independently. Discrepancies were discussed with or resolved by the last author.

### 2.2. Quality Assessment

The first author performed the quality assessment through Covidence, and the second checked two studies (20%) to ensure the quality and accuracy of the process. The quality appraisal of qualitative studies was performed with the critical appraisal skills program (CASP) qualitative checklist [21]. The checklist of the National Heart, Lung, and Blood Institute was selected to assess the quality of the cross-sectional and cohort studies [22]. This choice was made because no CASP checklist was available for cross-sectional studies at the time. A summary of the articles’ quality is available in Appendix A.

### 2.3. Data Analysis

The first author extracted the data using a standardised data collection form adapted from the ‘Data collection form for intervention reviews for RCTs and non-RCTs–template’ from the Cochrane Developmental, Psychosocial and Learning Problems Review Group [23]. The template included items that related to general information (on the study and data extraction), study eligibility, methods, participants, the phenomenon of interest, context, and other information (e.g., a study funding source, possible conflicts of interest, and conclusion). The second author checked 20% of the data extracted (2/10 studies) to confirm the quality of the data extraction.

Four corresponding authors of the included studies were contacted to clarify questions and attempted to retrieve additional details about their studies, such as the participant’s selection and sample calculation, among others. Three of them provided the requested details. When multiple records on the same study were merged in Covidence, the evidence was reported only once. Merging two records (e.g., an article and a thesis dissertation) on the same study was performed twice, providing more information on the methods used in the study.

As primary data were obtained from quantitative and qualitative studies, two separate narrative syntheses were performed. This method allowed us to synthesise the findings of diverse studies with heterogenous methodologies, which reported various factors that influenced the sustainability of donations to milk banks.

Overall, the evidence is reported with its origin and confidence level (very low, low, moderate, high). The evidence from the quantitative studies was assessed using the grading of recommendations, assessment, development, and evaluations (GRADE) assessment tool [24], and the evidence used qualitative studies with the GRADE confidence in the evidence from reviews of qualitative research (CERQual) assessment tool [25]. The GRADE tool enabled us to assess five domains: risk of bias, imprecision, inconsistency, indirectness, and publication bias. The GRADE-CERQual tool guided the assessment of the methodological limitations, coherence, adequacy, and relevance. Finally, an integration of the narrative syntheses from the quantitative and qualitative studies was conducted.

## 3. Results

A total of 6722 references (6720 studies) were identified. After the duplicates were removed, the titles and abstracts of 3012 studies (45%) were screened, and 2515 studies that were determined not to be relevant for this review were removed. The full texts of 497 studies (7%) were assessed for eligibility, and 487 studies were excluded. The excluded studies were mainly (1) not primary studies, (2) not on the phenomenon of interest, (3) published before 1990, or (4) had no full article available. Figure 1 describes the study selection and more details on the reasons for exclusion. Only 10 studies (0.15%) were included: eight quantitative observational studies (seven cross-sectional and one cohort studies) and two qualitative studies (one qualitative case study and one phenomenology research study) [26,27,28,29,30,31,32,33,34,35].

### 3.1. Characteristics of the Ten Selected Studies

All the studies were published in English between 2007 and 2021. They were conducted in five countries: Brazil, India, Italy, Spain, and the USA. Table 1 summarises their characteristics. The included studies took place in 12 HMBs, one health facility with a human milk donation service, and one health facility without one. The size of the studies varied; four studies involved fewer than 100 participants, three studies had between 100 and 500 participants, one study involved between 500 and 1000 participants, and two had more than 1000 participants.

Overall, the 10 studies had 7053 participants, among whom 6976 were donors, 44 were mothers who were either non-donor or potential donors, 18 were fathers, six were grandmothers, and nine were service providers. The most common variable for quantifying the donation sustainability (in seven studies) was the volume of milk donated [27,28,30,31,32,34,35]. Other variables used to quantify or qualify donation sustainability included facilitators and barriers to donation continuation [29], donation recurrence (first-time versus regular donors) [33], and the donors’ frequency of milk production and extraction [26] (Table 1).

### 3.2. Risk of Bias in the Included Studies

All the studies were assessed for risk of bias or confidence in the findings. Among the 10 studies, five (50%) had methodological limitations at different levels, such as a small sample size, unclear participant selection or recruitment, lack of sampling strategy, absence of explanation on how the sample size was calculated, or absence of control for confounding. Table 1 shows the overall evaluation of the methodological quality, with a rating of good for five studies [27,28,29,30,35], moderate for three studies [31,32,33], and weak for two studies [26,34]. Further details are provided in Appendix A.

A total of 30 factors that influenced the sustainability of the donation were identified in the literature. Of these, 13 were quantitative factors (Table 2), and 17 were qualitative factors (Table 3). The overall assessed confidence based on the GRADE and GRADE-CERQual was very low for the quantitative finding and low to very low for the qualitative findings. Additional details are provided in Table 2 and Table 3 and Appendix A.

### 3.3. Synthesis of Quantitative Findings

The domains that emerged from the quantitative findings (regarding the factors that influenced the sustainability of donation to HMBs) were factors related to the donation duration, donors’ infant features, and donors’ features (e.g., sociodemographic characteristics, motivation to donate, and milk donation history).

The volume of milk donated increased with the duration of the human milk donation [27,34]. The longer women donated their milk, the greater the volume they donated. Furthermore, one study found that women who began donating their milk earlier (before four months postpartum) tended to donate larger volumes than women who first donated after four months postpartum [35].

Among the investigated factors related to infants, the main finding was that women with preterm infants donated larger volumes of DHM than mothers with term infants. Two studies found that the mothers of infants who were <37 weeks donated significantly larger volumes of milk than mothers of (full) term infants [27,28]. One study found that mothers of infants who were <32 weeks donated more milk than mothers of infants of a greater gestational age [35]. Other converging findings were that mothers of infants with a low birthweight donated more DHM than mothers with higher birthweight infants, regardless of whether they were preterm or term [34].

Many factors that influenced the sustainability of donation in the included studies related to the donors themselves. A relationship between the number of pregnancies and donation recurrence was found in one Brazilian study. There was a lower likelihood of women with one to three pregnancies becoming regular donors (RR: 0.501, 95% CI: 0.286, 0.877) and a greater likelihood of donation recurrence among women with four to seven pregnancies (RR: 1.928, 95% CI: 1.039, 3.580) [33].

The same study also reported that the education level (primary, secondary, higher, or illiterate), whether complete or incomplete, could be related to donation recurrence. Women with a higher education level were twice as likely to become regular donors (RR: 2.062, 95% CI: 1.010, 4.213) [33].

Osbaldiston and Mingle [32] investigated the relationship between the donors’ motivation to donate and the volume of donations on a scale from 0 (strongly disagree) to 10 (strongly agree). Only two motivations were significantly related to the volume of donations and are, consequently, reported in this review. First, women who self-reported as ‘having an excess of milk and wanting to donate it’ donated greater volumes of milk (r = 0.31, *p* < 0.01) than women who disagreed with that statement. Second, donors who self-reportedly ‘needed to pump to stimulate lactation’ donated significantly more than those who disagreed with that statement (r = 0.34, *p* < 0.01). In another study, previous milk donors donated greater volumes (mean: 1.56 litre, 95% CI: 1.01, 2.11, *p* < 0.001) than first-time donors [35].

### 3.4. Synthesis of Qualitative Findings

The qualitative factors that were reported positively affect the frequency of milk production and extraction included the frequency of milk expression and the time of day when the mother pumped [26,29]. The factors with either a positive or negative impact on the frequency of milk production and extraction included the frequency of the infant’s feeding and the donor’s self-hydration, diet, and time available for milk extraction [26]. The factors reported to negatively influence the frequency of milk production and extraction included the baby’s growth and the donor’s fatigue, negative emotions (e.g., anger, anxiety), and activities (e.g., going out, using contraception, and returning to work) [26,29,30].

The donors reported three main obstacles to continued donation. First, they perceived an obstacle in the lack of support at their workplace (e.g., lack of emotional support or lack of encouragement to donate milk or breastfeed their own child). The other two barriers were the distance to the HMB (in the absence of human milk collection or transportation services) and that breastfeeding their own child could negatively impact the frequency of donations [29]. The same study highlighted that the support women received (from healthcare professionals and/or their families) positively impacted their willingness to continue donating. Finally, a shortage of human resources (including healthcare professionals) negatively affected the volume of donations [30].

### 3.5. Overall Synthesis of Quantitative and Qualitative Factors

The identified factors were triangulated and organised into categories that were congruent between the quantitative and qualitative findings, providing a more in-depth understanding of donation sustainability. The factors were synthesised into categories related to (1) donation duration, (2) features of the donors’ infants, (3) donors’ features (socio-demographic characteristics, motivation to donate, and milk donation history), and (4) factors related to the milk bank and healthcare system (awareness and support).

## 4. Discussion

To the best of our knowledge, this is the first systematic review to investigate the topic of the sustainability of human milk donation to HMBs and the factors that support women in continued milk donation.

Most of the factors influencing the sustainability of milk donation were micro-level factors related to the donors themselves or to their infants (summarised in Table 4). Very few factors were found at the meso level (milk bank), and none were found at the macro level (e.g., health care systems, policy, politics, or economic factors).


### 4.1. Implications for Practice

HMBs’ support of donation was associated with donors’ willingness to pursue milk donations (qualitative evidence). Despite the low to very low confidence level of this finding, this result suggests that the efforts of milk banks, healthcare services, and healthcare providers to support breastfeeding positively influence the sustainability of milk donation. It also highlights the importance of displaying information about milk extraction and donation.

One of the most studied factors among the included studies was the relationship between the duration of the donation and the volume of milk donated. While this seems self-evident, it is an important factor to highlight, as milk donation (unlike other biological material donations, e.g., blood) is temporally limited. Indeed, this relationship implies that early engagement with donors may support and promote longer donation durations. Interestingly, a study conducted in Europe found that 25% of the 123 participating European milk banks (of 281 existing) allowed women to donate only after a specific postpartum week [36].

Furthermore, the evidence suggests that recruiting mothers whose infants are of low gestational age or low birthweight may also support a larger volume of donations. A strategy to target and recruit donors who could be able to donate more of their milk might be cost-effective in terms of the resources allocated and increased availability of DHM for vulnerable newborns. The significant difference in the volume of donated milk between mothers of term infants and those of preterm infants is interesting but still little investigated. Future research should explore this further. The distance to the HMB was reported as a barrier to pursuing donations. Interesting options that merit being explored, such as developing milk collection and transport services, increasing the number of HMBs [29] (or milk depots) in some regions, and strengthening collaboration between HMBs. Finally, the finding that thrush in the infant is associated with larger volumes of donated milk may be difficult to explain and could be a confounding factor with regard to milk extraction.

### 4.2. Importance of Support and Resources

From an organisational perspective, the five factors #26–30 (Table 3) with low confidence (which is higher than the very low confidence level of the other factors) are mainly at the milk bank level. These factors are related to external support (from HMBs, family, and workplace) as well as to the logistics and human resources of HMBs. They seem to be factors that HMBs could potentially positively impact by promoting the need for donations and providing the resources to support them. The sustainability of donations could be facilitated by a deeper understanding of the details of these factors and of how HMBs could make a difference by providing information, supporting donors, and organising and adapting logistics (e.g., milk collection, milk transport, and adequate staff).

This review highlights the need for further research at the micro, meso, and macro levels to better understand the stakes regarding the sustainability of human milk donations to HMBs (not only at the donor level but also at the milk bank level) and the potential effect of policies, guidelines, and health care systems on HMB sustainability. From a public health perspective, such evidence is necessary to help adjust guidelines and national and regional policies. Further evidence is also needed to deepen the understanding and extend the impact of DHMs’ support and the promotion of donations on both the awareness and the sustainability of milk bank systems.

No studies have investigated DHM sustainability at the higher systemic (macro) level. Such research could provide data on the impact of recruitment strategies and on the sustainability and cost-effectiveness of donations. This may include national strategies to improve the sustainability of donations that could be adapted according to the local contexts of milk bank systems. Future studies should also identify the influence of guidelines, policies, and healthcare systems on the sustainability of milk bank systems at a more general level (e.g., financial resources and impact on the environment).

### 4.3. Limitations and Strengths

This review highlights some limitations of the existing studies that report factors influencing the sustainability of donations to HMBs. The included quantitative studies were cross-sectional (with the exception of one cohort study [28]), which does not allow us to infer causal relations. The weak to moderate quality of some studies represents another limitation. Future studies investigating these factors should take advantage of various study designs (including randomised controlled trials). Another limitation is that the present review investigated the sustainability of milk donation exclusively in the context of milk banking and ignoring milk sharing. However, this is also a strength, as the direct comparison between these two distinct contexts is difficult.

Among the limitations of the present review is the small number of studies found, in which very few factors (three of 30) were reported by more than one study. This review focused on the volume of milk donation but not on its quality (milk composition: term versus preterm milk for preterm infants), which is important but beyond the scope of this review. The present review includes studies from three high-income, one upper-middle, and one lower-middle-income country but from no low-income countries, which limits the generalisability and transferability of the findings. The concerned countries obviously differ in culture, economy, and healthcare systems, but they share an interesting commonality in their significant number of milk banks. Finally, in light of the paucity of studies on the sustainability of milk banking, the authors of this review reasoned that combining these results represented the best way to provide constructive input. It is hoped that future studies on this topic will soon enrich the literature, enabling more precise meta-synthesis or meta-analysis to support the tailored development of milk banking.

Beyond these acknowledged limitations, this review has several strengths. It is the first to investigate the sustainability of human milk donation and identifies several gaps in the literature. Moreover, the protocol was published in PROSPERO to ensure the transparency and rigor of the review. The risk of publication bias was limited by searching for grey literature in addition to the broad search of eight databases. The data collection and analyses were performed with the support of Covidence using high-quality and validated tools. The review was developed and inspired by Cochrane’s methods. Another strength is that the GRADE methods were employed to assess the quality of the evidence. Finally, this study joins others in calling for vulnerable infants’ equitable access to DHM [10] and highlights the need for more research on human milk donation and its sustainability [2].

## 5. Conclusions

This is the first known systematic review to investigate the factors impacting the sustainability of human milk donation to HMBs. The researchers assessed the studies’ quality, triangulated both qualitative and quantitative research studies, and synthesised the findings into four categories. In total, 30 factors that impacted the sustainability of human milk donation to HMBs were identified, and confidence in the evidence was assessed. Three categories were found at the micro level (donation duration, donors’ infants’ features, and donors’ features) and one at the meso level (factors related to milk banks and health care).

The findings suggest that the donors’ characteristics, their newborns’ health and factors related to HMBs, and their systems influence DHM donation volume and recurrence. Supporting and encouraging early donation and recruiting donors whose newborns are of low birth weight and gestational age may support the longer donation duration and larger volumes of donated milk. By informing the donor selection and retention processes, the identification of these factors will enable the development and evaluation of evidence-based strategies to encourage sustainable donations to HMBs that meet the needs of vulnerable infants.

More robust research is required to increase confidence in the existing findings on the sustainability of DHM donation to HMBs. Future studies are necessary to better understand the implications of the factors at a more systemic (macro) level, such as health care systems, strategies for donor recruitment, and their cost-effectiveness.

## Figures and Tables

**Figure 1 nutrients-14-05253-f001:**
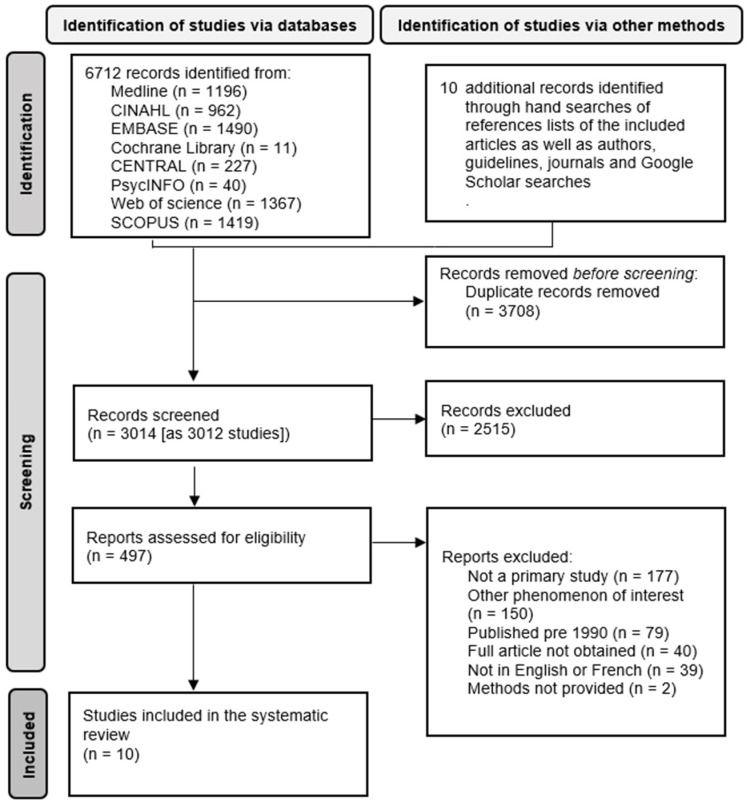
Flow diagram of the study selection.

**Table 1 nutrients-14-05253-t001:** Characteristics of the included studies (alphabetically by author).

Authors	Study Design	Sample Size, Sample (Population)	Setting (Context)	Country	Collection Period	Phenomenon of Interest	Sustainability Factors	Quality
Alencar and Seidl (2010) [26]	Cross-sectional	36 human milk donors	Two human milk banks of the public health system of the Federal District	Brazil	May 2005–November 2006	The categories for the reasons (factors)of influencing the frequency of expressing and milk production were the ingestion of liquids, diet, routines of the mother going out, contraceptive use and return to work, baby feeding frequency, the presence of negative emotions, availability of time, frequency of expressing, growth of the baby, period of the day, physical fatigue and laziness, and nothing interfering with production.	Donors’ frequency of milk extraction and milk production	Weak *
Bocci et al. (2019) [27]	Cross-sectional	304 human milk donors	A human milk bank in the province of Siena	Italy	January 2010–August 2017	Factors related to the volume of milk donated: length of donation period and gestational age (preterm delivery).	Volume of milk donated	Good *
Jarmoc et al. (2021) [28]	Cohort study	3764 human milk donors (with a total of 10,525 donations)	Mothers’ Milk Bank Northeast (MMBNE), a non-for-profit milk bank located in Newton Upper Falls, Massachusetts	USA	1 January 2011–1 September 2019	Factor related to the volume of milk donated: whether the mother had a preterm or a term infant.	Volume of milk donated	Good *
Machado et al. (2015) [29]	Qualitative	7 human milk donors	A human milk bank located in the Virgen de las Nieves hospital, Granada	Spain	May–June 2013	Factors that influence women to continue donating their milk: milk bank support and family support of donation.Obstacles for women to remain donors: distance to the milk bank, support at work, and reduction in milk by the process of breastfeeding itself.	Facilitators and barriers to donation continuation	Good **
Mondkar et al. (2018) [30]	Qualitative	56 service recipients, including human milk donors and key influencers, as well as 9 service providers	Two health facilities under the Mumbai Municipal Corporation were purposively selected as case studies: a Level III facility with a human milk bank and a Level II facility without one.	India	August–November 2016	service providers reported a factor that could be a barrier to sustaining an adequate supply of DHM: staff shortage.	Volume of milk donated	Good **
Nangia et al. (2020) [31]	Cross-sectional	1553 human milk donors	The human milk bank of a tertiary care centre in a low- and middle-income country	India	7 June 2017–28 February 2019	The volume of the donor milk collected is influenced by the service in which the baby is hospitalized (neonatal intensive care unit versus postnatal care ward).	Volume of milk donated	Moderate *
Osbaldiston & Mingle (2007) [32]	Cross-sectional	87 human milk donors and 19 non donors (women who had pumped milk while breastfeeding their infants).	Mothers’ milk bank at Austin (Texas)	USA	Fall 2005–spring 2006	Factors of interest apparently related to the amount of milk donated: thrush (in the infant); motive for donating milk (e.g., ’had too much milk and wanted to donate it’, as well as ‘needed to pump milk to stimulate lactation’); mother’s age.	Volume of milk donated	Moderate *
Pimenteira Thomaz et al. (2008) [33]	Cross-sectional	737 human milk donors	All three human milk banks in the State of Alagoas (located in the following hospitals: Maternity School Hospital Santa Mônica; Federal University of Alagoas School Hospital; Santa Casa de Misericórdia of São Miguel dos Campos).	Brazil	March 2004–February 2005	The most common characteristics of a regular donor in comparison with a first-time donor are having between four and seven pregnancies and possessing a higher education level.	Donation recurrence (first-time versus regular donors)	Moderate *
Quitadamo et al. (2018) [34]	Cross-sectional	90 women, enrolled for donation	The human milk bank of the Casa Sollievo della Sofferenza hospital, which is integrated into the Neonatology Service of the same hospital.	Italy	1 January 2014–31 December 2015	Factors related to the volume of milk donated: maternal age; birth weight of the neonate; duration of donation; profession of the donor.	Volume of milk donated	Weak *
Sierra-Colomina et al. (2014) [35]	Cross-sectional	391 human milk donors (for a total of 415 donations)	The human milk bank of the Hospital Doce de Octubre, in the Madrid community, Spanish central region.	Spain	1 January 2009–31 April 2013	Donors’ social and demographic variables related to the volume of donor milk delivered: previous donors; smaller gestational age of children; started donation at earlier stages of lactation.	Volume of milk donated	Good *

*: Quality appraisal of cross-sectional and cohort studies performed with the National Heart, Lung, and Blood Institute checklist. **: Quality appraisal of qualitative studies performed with the CASP qualitative checklist.

**Table 2 nutrients-14-05253-t002:** Summary of quantitative findings (factors influencing the sustainability of DHM donation to HMB) and GRADE assessment.

13 Factors	Summary of Findings	Dependent Variable (Sustainability Factor)	Reported Impact on Donation Sustainability: Increases + Decreases −	Studies Contributing to the Review	GRADE Assessment of Confidence in The Evidence *	Explanation of GRADE Assessment
FACTORS RELATED TO DONATION DURATION						
#1 Duration of donation	The duration of the donation is positively associated with the volume of milk donated. The longer women donate their milk, the larger volume they donate.	Volume of milk donated	+	Bocci et al., 2019 Quitadamo et al., 2018 [27,34]	Very low confidence ⊕○○○	Risk of bias not serious: 1 study with low risk of bias (good methodology) and 1 with very serious concerns. Unexplained inconsistency. No serious indirectness. Serious imprecisions. Possible publication bias suspected.
#2 Start of donation	Women who started donating their milk sooner (before 4 months postpartum) donated larger volumes in total than women who started donating later (≥4 months postpartum).	Volume of milk donated	+	Sierra-Colomina et al., 2014 [35]	Very low confidence ⊕○○○	Risk of bias not serious: 1 study with low risk of bias. Only 1 study where we were unable to assess inconsistency. No serious indirectness. Serious imprecision. Possible publication bias suspected.
FACTORS RELATED TO DONORS’ INFANTS FEATURES						
#3 Preterm infants (gestational age)	Women with a preterm infant donated larger volumes than mothers of term infants.	Volume of milk donated	+	Bocci et al., 2019 Jarmoc et al., 2021 Sierra-Colomina et al., 2014 [27,28,35]	Very low confidence ⊕○○○	Risk of bias not serious: 3 studies with low risk of bias (good methodology). Inconsistency not explained. No serious indirectness. Serious imprecision between 3 studies. Possible publication bias suspected.
#4 Birthweight	Infant birthweight correlated negatively with the volume of milk donated by the mother; mothers of infants with low birth weight donated more human milk.	Volume of milk donated	−	Quitadamo et al., 2018 [34]	Very low confidence ⊕○○○	Very serious limitations regarding risk of bias:1 study with weak methodology. Only 1 study where we were unable to assess inconsistency. No serious indirectness. Serious imprecisions. Possible publication bias suspected.
#5 Admission to the Neonatal Intensive Care Unit (NICU)	Admission of neonates to the NICU was associated with larger donation volumes by mothers.	Volume of milk donated	+	Nangia et al., 2020 [31]	Very low confidence ⊕○○○	Serious limitations due to risk of bias: 1 study with moderate risk of bias. Inconsistency not explained. No serious indirectness. Very serious imprecisions. Possible publication bias suspected.
#6 Thrush (in the infant)	Mothers who reported that their infants had thrush donated more milk than mothers who reported that their infant did not.	Volume of milk donated	+	Osbaldiston and Mingle, 2007 [32]	Very low confidence ⊕○○○	Serious limitations due to the study’s moderate risk of bias. Only 1 study where we were unable to assess inconsistency. No serious indirectness. Serious imprecisions: only 1 study. Possible publication bias suspected.
FACTORS RELATED TO DONORS’ FEATURES: Socio demographic characteristics						
#7 Maternal age	Two independent studies investigated the potential influence of maternal age on the volume of donation and yielded diverging results. Maternal age was both positively (Quitadamo et al. 2018) and negatively (Osbaldiston and Mingle, 2007) associated with the volume of donation.	Volume of milk donated	+/−	Osbaldiston and Mingle, 2007 Quitadamo et al., 2018 [32,34]	Very low confidence ⊕○○○	Serious limitations: 1 study with moderate risk of bias and 1 study with high risk of bias. Inconsistency not explained. No serious indirectness. Serious imprecision. Possible publication bias suspected.
#8 Number of pregnancies	Women with 4 to 7 pregnancies have a higher likelihood of donation recurrence (donating their milk more than once) than women with between 1 and 3 pregnancies.	Donation recurrence (first-time versus regular donors)	+	Pimenteira Thomaz et al., 2008 [33]	Very low confidence ⊕○○○	Serious limitations: 1 study with moderate risk of bias. Only 1 study where we were unable to assess inconsistency). No serious indirectness. Serious imprecisions (only 1 study). Possible publication bias suspected.
#9 Education level	Women with a higher education level had a greater likelihood of donation recurrence.	Donation recurrence (first-time versus regular donors)	+	Pimenteira Thomaz et al., 2008 [33]	Very low confidence ⊕○○○	Serious limitations due to 1 study with moderate risk of bias. Only 1 study where we were unable to assess inconsistency). No serious indirectness. Serious imprecisions (only 1 study. Possible publication bias suspected.
#10 Profession	Women who were unemployed, homemakers, or workers donated significantly smaller volumes of milk than women in other professional categories.	Volume of milk donated	+/−	Quitadamo et al., 2018 [34]	Very low confidence ⊕○○○	Very serious limitations due to weak methodology. Only 1 study where we were unable to assess inconsistency. No serious indirectness. Serious imprecisions (only 1 study). Possible publication bias suspected.
FACTORS RELATED TO DONORS’ FEATURES: Motivation to donate						
#11 Excess of milk	When donors were asked about their motivations to donate, one variable was ‘Having an excess of milk‘ (on a 0–10 scale from ‘not at all for this reason‘ to ’very much for this reason‘). A correlation was found between this variable and the volume of milk donated. Women who self-reported donating because of an excess of milk donated larger volumes than women who did not donate for that reason.	Volume of milk donated	+	Osbaldiston & Mingle, 2007 [32]	Very low confidence ⊕○○○	Serious limitations due to the study’s moderate risk of bias. Only 1 study where we were unable to assess inconsistency. No serious indirectness. Serious imprecisions (only 1 study). Possible publication bias suspected.
#12 Pumping to stimulate lactation	When donors were asked about their motivations to donate, one variable was ‘needed to pump to stimulate lactation‘ (on a 0–10 scale from ’not at all for this reason‘ to ’very much for this reason‘). Donors who reported donating for this reason (answers 7–10 on the scale) donated larger volumes than donors who did not donate for this reason (answer 0 on the scale).	Volume of milk donated	+	Osbaldiston and Mingle, 2007 [32]	Very low confidence ⊕○○○	Serious limitations due to the study’s moderate risk of bias. Only 1 study (unable to assess inconsistency). Not serious indirectness. Serious imprecisions (only 1 study). Possible publication bias suspected.
FACTORS RELATED TO DONORS’ FEATURES: Milk donation history						
#13 Previous milk donation	Donors who had previously been donors donated significantly greater volumes of milk than women who had not previously donated.	Volume of milk donated	+	Sierra-Colomina et al., 2014 [35]	Very low confidence ⊕○○○	No serious risk of bias due to the study’s good methodology. Only 1 study where we were unable to assess inconsistency. No serious indirectness. Serious imprecision (only 1 study). Possible publication bias suspected.

* The confidence level scale ranges from: ⊕○○○ very low confidence.

**Table 3 nutrients-14-05253-t003:** Summary of qualitative findings (factors influencing sustainability of DHM donation to HMB) and CERQual assessment.

17 Factors	Summary of Review Finding	Sustainability Factors	Reported Impact on Donation Sustainability: Increases + Decreases −	Studies Contributing to the Review	Study Design	CERQual Assessment of Confidence in the Evidence *	Explanation of CERQual Assessment
FACTORS RELATED TO DONORS’ INFANTS’ FEATURES							
#14 Baby feeding frequency	The frequency of the baby’s feeding was self-reported by some donors as potentially having a positive or negative influence on the frequency of milk extraction and milk production.	Frequency of milk Extraction and milk production	+/−	Alencar and Seidl, 2010 [26]	Quantitative	Very low confidence ⊕○○○	One study with no or very minor concerns about coherence, moderate concerns regarding relevance, and serious concerns regarding adequacy and methodological limitations.
#15 Growth of the baby	The baby’s growth was self-reported by some donors as potentially having a negative influence on the frequency of milk extraction and production.	Frequency of milk extraction and milk production	−	Alencar and Seidl, 2010 [26]	Quantitative	Very low confidence ⊕○○○	One study with no or very minor concerns about coherence, moderate concerns regarding relevance, and serious concerns about methodological limitations and adequacy.
FACTORS RELATED TO DONORS’ FEATURES: Health							
#16 Self-hydration	Self-hydration was self-reported by donors as potentially having a negative and/or positive influence on the frequency of milk extraction and milk production.	Frequency of milk extraction and milk production	+/−	Alencar and Seidl, 2010 [26]	Quantitative	Very low confidence⊕○○○	One study with serious concerns about methodological limitations. No or very minor concerns about coherence. Thin data from 1 country. Serious concerns regarding adequacy and moderate concerns regarding relevance.
#17 Diet	Diet was self-reported by donors as having potentially a negative and/or positive influence on the frequency of milk extraction and production.	Frequency of milk extraction and milk production	+/−	Alencar and Seidl, 2010 [26]	Quantitative	Very low confidence ⊕○○○	One study with serious concerns about methodological limitations. No or very minor concerns about coherence. Thin data from 1 country. Serious concerns regarding adequacy and moderate concerns regarding relevance.
#18 Physical fatigue	Fatigue was self-reported by donors as potentially having a negative influence on the frequency of milk extraction and production.	Frequency of milk extraction and milk production	−	Alencar and Seidl, 2010 [26]	Quantitative	Very low confidence ⊕○○○	One study with serious concerns about methodological limitations. No or very minor concerns about coherence. Thin data from 1 country. Serious concerns regarding adequacy and moderate concerns regarding relevance.
#19 Presence of negative emotions	Some donors self-reported that the presence of negative emotions could negatively influence the frequency of milk extraction and production.	Frequency of milk extraction and milk production	−	Alencar and Seidl, 2010 [26]	Quantitative	Very low confidence ⊕○○○	One study with serious concerns about methodological limitations. No or very minor concerns about coherence. Thin data from 1 country. Serious concerns regarding adequacy and moderate concerns regarding relevance.
FACTORS RELATED TO DONOR’S FEATURES: Motivation to donate							
#20 Availability of time	Donors self-reported that the availability of time (to pump) could negatively or positively influence the frequency of milk extraction and production.	Frequency of milk extraction and milk production	+/−	Alencar and Seidl, 2010 [26]	Quantitative	Very low confidence ⊕○○○	One study with serious concerns about methodological limitations. No or very minor concerns about coherence. Thin data from 1 country. Serious concerns regarding adequacy and moderate concerns regarding relevance.
FACTORS RELATED TO DONORS’ FEATURES: Breastfeeding and milk expression							
#21 Experience of breastfeeding simultaneously	Donors reported a decrease in donation frequency as their milk production decreased due to the process of breastfeeding itself (less excess than previously).	Obstacle to remaining a donor: frequency of donation	−	Machado et al., 2015 [29]	Qualitative	Low confidence ⊕⊕○○	One study with minor concerns about methodological limitations. No or very minor concerns about coherence. Thin data from 1 country. Serious concerns regarding adequacy and moderate concerns regarding relevance.
#22 Frequency of milk expression	Some donors self-reported that the frequency of milk expression had a potentially positive influence on the frequency of milk extraction and milk production.	Frequency of milk extraction and milk production	+	Alencar and Seidl, 2010 [26]	Quantitative	Very low confidence ⊕○○○	One study with serious concerns about methodological limitations. No or very minor concerns about coherence. Thin data from 1 country, serious concerns regarding adequacy, and moderate concerns regarding relevance.
#23 Nothing interferes with milk production	Some donors self-reported that nothing interfered with milk production, which was seen as a positive influence on the frequency of milk extraction and production.	Frequency of milk extraction and milk production	+	Alencar and Seidl, 2010 [26]	Quantitative	Very low confidence ⊕○○○	One study with serious concerns about methodological limitations. No or very minor concerns about coherence. Thin data from 1 country. Serious concerns regarding adequacy and moderate concerns regarding relevance.
#24 Time of day for expressing milk	Donors self-reported that the time of day possibly influenced the frequency of milk extraction and production (increased production at night)	Frequency of milk extraction and milk production	+	Alencar and Seidl, 2010 [26]	Quantitative	Very low confidence ⊕○○○	One study with serious concerns about methodological limitations. No or very minor concerns about coherence. Thin data from 1 country. Serious concerns regarding adequacy and moderate concerns regarding relevance.
FACTORS RELATED TO DONORS’ FEATURES: Other							
#25 Mother’s routines (going out, contraceptive use, return to work)	Some donors self-reported that going out, using contraception, or returning to work may have negatively influenced the frequency of milk extraction and production.	Frequency of milk extraction and milk production	−	Alencar and Seidl, 2010 [26]	Quantitative	Very low confidence ⊕○○○	One study with serious concerns about methodological limitations. No or very minor concerns about coherence. Thin data from 1 country. Serious concerns regarding adequacy and moderate concerns regarding relevance.
MILK BANK and HEALTH CARE–RELATED FACTORS: Support							
#26 Milk bank support to donation	The mother’s environment (support offered by MB staff) had a positive influence on her willingness to continue donating.	Donor willingness to continue donating milk	+	Machado et al., 2015 [29]	Qualitative	Low confidence ⊕⊕○○	One study with minor concerns about methodological limitations. No or very minor concerns about coherence. Thin data from 1 country. Serious concerns regarding adequacy and moderate concerns regarding relevance.
#27 Family support to donation	The support mothers received from their family positively influenced their willingness to continue donating.	Donor willingness to continue donating milk	+	Machado et al., 2015 [29]	Qualitative	Low confidence ⊕⊕○○	One study with minor concerns about methodological limitations. No or very minor concerns about coherence. Thin data from 1 country. Serious concerns regarding adequacy and moderate concerns regarding relevance.
#28 Work impact and support	Donors reported that incomprehension and lack of support at their workplace was an obstacle to remaining a donor.	Obstacle to remaining a donor	−	Machado et al., 2015 [29]	Qualitative	Low confidence ⊕⊕○○	One study with minor concerns about methodological limitations. No or very minor concerns about coherence. Thin data from 1 country. Serious concerns regarding adequacy and moderate concerns regarding relevance.
MILK BANK and HEALTH CARE–RELATED FACTORS: Logistics							
#29 Distance from milk bank	Donors reported that the distance they had to travel to deliver their milk to the milk bank (no home collection service being available) was an obstacle to becoming and remaining a milk donor.	Obstacle to remaining a donor	−	Machado et al., 2015 [29]	Qualitative	Low confidence ⊕⊕○○	One study with minor concerns about methodological limitations. No or very minor concerns about coherence. Thin data from 1 country. Serious concerns regarding adequacy and moderate concerns regarding relevance.
#30 Human resources	Health care providers felt that a shortage of human resources in milk banks negatively affected the volume of the milk collected.	Volume of milk donated	−	Mondkar et al., 2018 [30]	Qualitative	Low confidence ⊕⊕○○	One study with minor concerns about methodological limitations. No or very minor concerns about coherence. Thin data from 1 country. Serious concerns regarding adequacy and moderate concerns regarding relevance.

* The confidence level scale ranges from: ⊕○○○ very low confidence; ⊕⊕○○ low confidence.

**Table 4 nutrients-14-05253-t004:** List of factors and reported impacts on donation sustainability at the micro, meso, and macro levels.

30 Factors (at the Micro, Meso, and Macro Levels)	Reported Impact on Donation Sustainability *	Study Design and Reference (Cross Sectional Design Unless Otherwise Noted)
Micro Level (Donors, Their Infants and Families)		
□ Duration of donation	+	[27,34]
□ Start of donation	+	[35]
□ Preterm infant (gestational age)	+	Cross sectional & cohort study [27,28,35]
□ Birth weight	−	[34]
□ Admission to NICU	+	[31]
□ Thrush in the infant	+	[26]
□ Baby feeding frequency	+/−	[26]
□ Growth of the baby	−	[32]
□ Maternal age	+/−	[32,34]
□ Number of pregnancies	+	[33]
□ Education level	+	[33]
□ Profession	+/−	[34]
□ Self-hydration	+/−	[26]
□ Diet	+/−	[26]
□ Physical fatigue	−	[26]
□ Presence of negative emotions	−	[26]
□ Motivated to donate by having an excess of milk	+	[32]
□ Motivated to donate by having to pump to stimulate lactation	+	[32]
□ Donor’s availability of time	+/−	[26]
□ Experience of breastfeeding simultaneously	−	Qualitative [29]
□ Frequency of milk expression	+	[26]
□ Nothing interferes with milk production	+	[26]
□ Time of day for expressing milk	+	[26]
□ Family support for donation	+	Qualitative [29]
□ Mother’ routines (going out, contraceptive use, return to work)	−	[26]
□ Previous milk donation history	+	[35]
Meso level (milk bank level)		
□ Milk bank support to donation	+	Qualitative [29]
□ Work impact and support	−	Qualitative [29]
□ Distance to milk bank	−	Qualitative [29]
□ Human resources of HMB	−	Qualitative [30]
Macro level (system)		
None reported		

* (+) increases; (−) decreases.

## Data Availability

The data used to support the findings of this review are available from the corresponding author upon request.

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
