# Peer review of "Factors That Influence the Sustainability of Human Milk Donation to Milk Banks: A Systematic Review"

_nutrients, 2022, doi:10.3390/nu14245253_

Round 1
Reviewer 1 Report
The importance of human milk for preterm infants is well known. The authors address a rather ill-considered question - how to ensure sustainable supplies of human milk to donor human milk banks.
Generally, I thought the paper well written and timely. It's main limitation is the very low to low confidence around most of the identified factors.
My suggestions are minor ones:
1. Introduction
> I think the cost-benefits the authors describe are fanciful and use extremely poor methodologies. When proper methods are used (e.g. Pediatrics (2018) 141 (3): e20170737) the cost savings disappear. The reason to reduce the rate of NEC in preterm babies is to reduce the rate of NEC!
2. Methods
> These seem fairly standard, and appropriate for a systematic review
3. Results
> The studies come from countries with very different cultures, and economic conditions, and health-care systems. Is it appropriate to combine them into a single analysis?
> What precise criteria were used to grade study quality (Table 1)?
> Could the difference between mothers of term and preterm babies be that preterm infants took less milk that term infants? Or that mothers of preterm infants were only permitted to donate if they had very large supplies of milk?
> Did education level affect the success of being a donor, or the accessibility in becoming a donor in the first place?
> Table 2b. Factor #16 is missing the visual representation of confidence
> Factors #26 - #30 seem to have the highest confidence associated with them. They are also share similarities, being related to external/ societal support. They are also modifiable. They should be highlighted as a theme in the discussion.
> I though Table 3 was a very helpful presentation of the factors.
Discussion
> It should perhaps be mentioned that the outcome of quantity of donated milk is different to that of quality of donated milk. It is possible that in some cases they a re in opposition (protein content) while in others they may not be (effect on NEC).
Reviewer 2 Report
This review aimed to identify factors impacting the sustainability of human milk donation to milk banks. The review advances knowledge on factors that promote the sustainability and duration of milk donations to HMBs. It is the first review to investigate the sustainability of human milk donation. Those results are referential to related researchers. The paper has clear logic and complete framework, but needs some improvement before further consideration for publication. Some detailed comments are as follows:
1. On Table 1: This table described 10 selected studies, some sustainability factors are the same, which should be further summarized and grouped according to the same sustainability factors. This will be looked more organized.
2. On section 2.1, the content of search strategy and selection criteria should be simplified and concise. There is no need to go into so much detail about a collection and screening.
3. The format of the references also needs to be double-checked. For example, references such as the ninth and twentieth are cited in an incorrect format.
4. On section 3.1, this section mainly summarizes the content of the 10 studies, the title of section can be changed to the characteristics of the ten selected studies.
